# Wildfires enhance phytoplankton production in tropical oceans

Dongyan Liu [1✉], Chongran Zhou [1], John K. Keesing [2✉], Oscar Serrano [3,4], Axel Werner [3], Yin Fang [5], Yingjun Chen [6✉], Pere Masque [3,7,8], Janine Kinloch [9], Aleksey Sadekov [10] & Yan Du [11]

Wildfire magnitude and frequency have greatly escalated on a global scale. Wildfire products rich in biogenic elements can enter the ocean through atmospheric and river inputs, but their contribution to marine phytoplankton production is poorly understood. Here, using geochemical paleo-reconstructions, a century-long relationship between wildfire magnitude and marine phytoplankton production is established in a fire-prone region of Kimberley coast, Australia. A positive correlation is identified between wildfire and phytoplankton production on a decadal scale. The importance of wildfire on marine phytoplankton production is statistically higher than that of tropical cyclones and rainfall, when strong El Niño Southern Oscillation coincides with the positive phase of Indian Ocean Dipole. Interdecadal chlorophyll-a variation along the Kimberley coast validates the spatial connection of this phenomenon. Findings from this study suggest that the role of additional nutrients from wildfires has to be considered when projecting impacts of global warming on marine phytoplankton production.

[1] State Key Laboratory of Estuarine and Coastal Research, Institute of Eco-Chongming, East China Normal University, Shanghai 200062, China. [2] CSIRO Oceans and Atmosphere Research, and University of Western Australia Oceans Institute, Indian Ocean Marine Research Centre, Crawley, WA, Australia. [3] School of Science and Centre for Marine Ecosystems Research, Edith Cowan University, Joondalup, WA, Australia. [4] Centro de Estudios Avanzados de Blanes, Consejo Superior de Investigaciones Científicas, Blanes, Spain. [5] College of Marine Ecology and Environment, Shanghai Ocean University, Shanghai 201306, China. [6] Shanghai Key Laboratory of Atmospheric Particle Pollution and Prevention, Department of Environmental Science and Engineering, Fudan University, Shanghai 200438, China. [7] Departament de Física and Institut de Ciència i Tecnologia Ambientals, Universitat Autònoma de Barcelona, Bellaterra, Spain. [8] International Atomic Energy Agency, 4a Quai Antoine 1er, 98000 Principality of Monaco, Monaco. [9] Biodiversity and Conservation Science, Department of Biodiversity, Conservation and Attractions, Bentley Delivery Centre, Bentley, WA, Australia. [10] Ocean Graduate School, ARC Centre of Excellence for Coral Reef Studies, University of Western Australia, Crawley, WA 6009, Australia. [11] State Key Laboratory of Tropical Oceanography, South China Sea Institute of Oceanology, Chinese Academy of Sciences, Guangzhou, China. ✉email: dyliu@sklec.ecnu.edu.cn; john.keesing@csiro.au; yjchenfd@fudan.edu.cn

Marine phytoplanktons are critical primary producers, contributing nearly half of the biosphere's net primary production[1]. The impact of global warming on marine phytoplankton production (MPP) has become increasingly evident in the past few decades[2], but how MPP is affected at different latitudes remains controversial[3–5]. For example, assimilating ocean colour satellite data into a marine biogeochemical model showed that global net MPP experienced a small yet significant decline of $-0.8$ Pg C yr$^{-1}$ or $-2.1\%$ decade$^{-1}$ from 1998 to 2015[4]. Ocean stratification enhanced by warming has been invoked as a major mechanism to explain the decline in MPP: warming modifies the mixed layer depth and reduces the vertical mixing of the surface layer and underlying cooler nutrient-rich waters below the permanent pycnocline[6]. Changes in these physical processes lead to a reduced supply of nutrients to the upper ocean and, consequently, have a negative impact on MPP; for example, large-sized phytoplankton species, such as diatoms, can significantly decrease in abundance and cause MPP decline in the upper ocean[7,8]. However, although this might be true in temperate oceans, it is not always the case in the tropics and Polar Regions[9,10]. Generally, the rise in temperature and changes in stratification in tropical oceans are smaller than in temperate oceans because of their larger heat capacity and thermal inertia[9]. Moreover, warming-induced secondary climate effects, such as increased tropical cyclones and upwellings, can compensate for nutrient depletion in the upper ocean by accelerating turbulent mixing and promoting rainfall[11]. Phytoplankton metabolic capacity in the tropics is also higher than that at high latitudes[12]. These factors can partially offset the negative effect of greater stratification and even lead to a stage increase in MPP in tropical oceans[5,11].

In contrast to the impact of physical processes on MPP in tropical oceans, the role of wildfires has received minimal attention. The risk and severity of wildfires in the southern hemisphere have greatly escalated on a global scale as a consequence of rising temperatures and more frequent heat waves[13,14]; for example, there was a distinct lengthening of the fire weather season between 1980 and 2018 (Fig. 1a). The emissions and ash from wildfire are rich in biogenic elements[15], such as nitrogen, phosphate, silicate, and iron, and exert a distinct impact on atmospheric and aquatic environments[16,17]. For example, particles emitted by wildfires account for approximately 62% of the global annual emissions of organic matter from biomass burning[16]. The global flux of soluble charcoal from biomass burning is estimated to be 40–250 million tons yr$^{-1}$, and

approximately 26.5 million tons enter the ocean every year[17]. Despite their importance, our understanding of the effect of wildfires on the ocean is far less than our understanding of their role in terrestrial ecosystems.

Northern Australia is one of the most fire-prone savanna regions of the world (Fig. 1a). As with most of Australia, hot weather associated with the El Niño Southern Oscillation (ENSO) is the main driver of wildfires[18]; the risk and severity of wildfires can increase when the Indian Ocean Dipole (IOD) is also in its positive phase (pIOD). The drying effect of the easterly shift in equatorial trade winds induced by pIOD can promote fire conditions and lead to extra-strong wildfires[19]. For instance, during the 2019–2020 bushfires, a globally unprecedented 21% of the Australian temperate and broadleaf mixed forest biome burned[18,19]. Hypothetical conjecture, when strong ENSO occurs during the pIOD phase, escalating wildfires can increase the flux of biogenic elements into the ocean via the pathways of atmospheric deposition and riverine input and lead to higher MPP. To prove their connection in the context of climate modes, it is necessary to examine a decadal relationship between wildfire magnitude and MPP.

Palaeoecological methods can provide an effective pathway to reconstruct the interdecadal correlations between environmental and biological information. Total organic carbon (TOC), total nitrogen (TN), and biogenic silicate (BSi) preserved in sediment cores are important geochemical proxies to reconstruct MPP in the upper ocean. TOC and TN reflect the accumulation of organic matter in the seafloor, while BSi is a frustule component of diatoms and is often used as a proxy for diatom biomass in the ocean because of its major contribution to MPP[20,21]. The effectiveness of BSi as an indicator of MPP in northern Australia has been validated using biomarkers[22]. Wildfire magnitude can be reconstructed using black carbon (BC) contents preserved in the sediments. BC is an organic, molecularly diverse product resulting from the incomplete combustion of biomass and fossil fuels[23] and it decomposes slowly after burial in marine sediments as a component of TOC[17]. Moreover, the source of BC from biomass or fossil fuels can be distinguished via the ratios char/soot. In high-temperature fossil fuel combustion (e.g. vehicle emissions and industrial coal combustion) the ratio of char/soot ratio is less than one, while for the relatively low-temperature biomass burning (e.g. wildfire), the ratio of char/soot is much higher than one[24].

Therefore, we measured these geochemical parameters in three sediment cores (ID: 185, 200, and KGR) collected from the

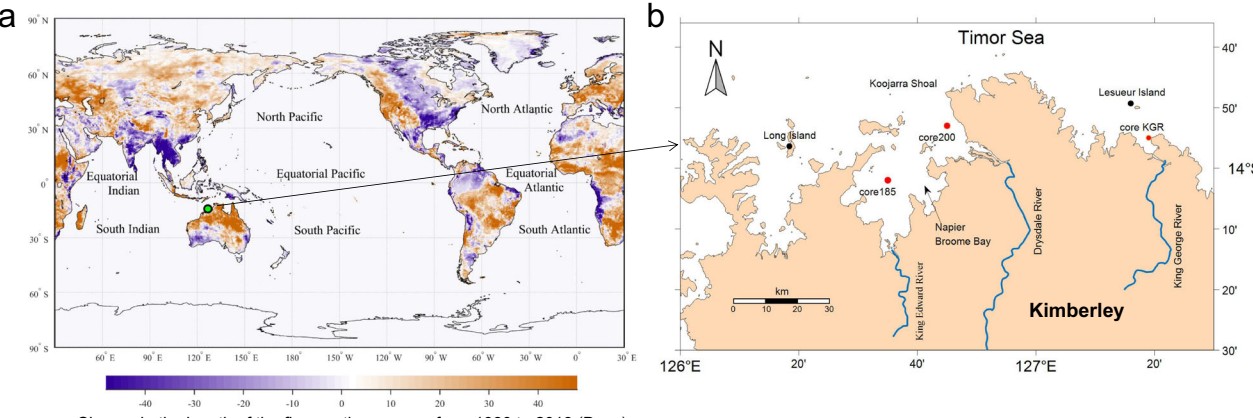

**Fig. 1 Maps showing changes in the length of the fire weather season on a global scale and sampling location. a** Map showing the change in the length of the fire weather season between 1980 and 2018 (data from the ERA5 data set[13,14]; the green circle indicates the Kimberley coast, a fire-prone region in northern Australia). **b** Sampling sites of core 185, core 200, and core KGR (red dots) in the Kimberley coast.

Kimberley coast of northern Australia (Fig. 1b), aiming to reconstruct a 100-year time-series relationship between wildfire magnitude and MPP variation. Kimberley region is one of the most fire-prone savanna regions of the world (Fig. 1a). The Kimberley coastline and its river catchments are sparsely populated and remote, with relatively low levels of fuel consumption[25]. The ratios char/soot in the three cores ranged 6.9 to 22.3 for core 185, from 9.8 to 26.2 for core 200, and from 12.3 to 31.7 for core KGR, respectively (Supplementary Fig. 1), these numbers indicate that the BC in marine sediments of Kimberley coast predominantly originates from wildfires. Thus, it is reasonable to reconstruct a chronological relationship between wildfire magnitude and MPP, using BC and BSi, respectively. The correlation between BC and BSi and their characteristics over time are analysed in the context of IOD and ENSO climate modes (Fig. 2a), referring to the fire record of the National Oceanic and Atmospheric Administration (NOAA) during 1988–2018 and the variation of satellite chlorophyll-a (Chl-a) of Kimberley coast from 2003 to 2018. The importance of wildfire effect on phytoplankton production is statistically compared with that of tropical cyclones and rainfall during pIOD and negative IOD (nIOD) phases, respectively.

## Results

**The positive effect of wildfire on MPP**. The period between 1920 and 2017 can be divided into three climate modes, including nIOD phase (1920–1960), a fluctuation between nIOD and pIOD phases (1960–2005), and pIOD phase (2005–2018) (Fig. 2a). The three sedimentary cores (185, 200, and KGR) cover the time span from the 1920s to 2010s, representing two phases of pIOD dominance (1991–2017) and nIOD dominance (1926–1990). There are two common characteristics in the three cores: (1) BSi and BC displayed a strongly positive correlation in the three cores during the phase of pIOD dominance (Table 1). (2) The frequency and magnitude of RSI increased evidently in the three cores during the phase of pIOD dominance (Fig. 2b–g); a common increase for TOC, TN, BSi, and BC appeared after 2010s when strong ENSO conditions coincide with pIOD phase (Fig. 2). These positive signals indicate an important linkage between wildfire and MPP during pIOD phase.

During the phase of nIOD dominance (1926–1990), no positive correlation was found between TOC, TN, BSi, and BC in Core 185 and KGR, but BC displayed positive correlations with TOC ($r = 0.56$, $p < 0.01$) and TN ($r = 0.56$, $p < 0.05$) in Core 200 (Table 1). During the phase of pIOD dominance (1991–2017), BSi and BC displayed strongly positive correlations with TOC and TN in Cores 200, a positive correlation between BC and TOC ($r = 0.67$, $p < 0.05$) was found in Core KGR (Table 1). In comparison, the positive contribution of BSi and BC to organic matter greatly increased during the phase of pIOD dominance. After the 1990s, the contents of BSi and BC in the three cores changed in a V-shape (Fig. 2c, e, g). These two peaks are basically consistent with the time when strong ENSO (2003, 2015) overlapped with pIOD (Fig. 2a). The synchronous variations between BSi and BC in the pIOD phase indicate the impact of wildfires.

The contents of TOC, TN, BSi, and BC in the three cores are different. In comparison, they were higher in Core185 than in Core 200 and KGR (Fig. 2b–g). This might be related to the difference of grain sizes in the three cores and the variations of freshwater discharge from the three rivers or differences in catchment size or characteristics such as size soil and vegetation type (Supplementary Fig. 2), e.g. the median grain size ($d_{50}$) in Core 185 showed a distinct decrease during 1998–2018, corresponding to relatively lower BC (Fig. 2c); in contrast, $d_{50}$

in Core 200 showed a distinct increase during the 1990s, corresponding to relatively higher BC contents. However, these differences in concentrations did not affect the positive correlation between BSi and BC during pIOD phase. These results are consistent with the findings in northern Australia, where there was a 1.5–3 times increase in primary production between the 1960s and 2010s, mainly contributed by diatoms and high primary production occurred during the ENSO years[22].

**Observational data to verify palaeo-reconstruction**. To verify the accuracy of palaeo-reconstruction using geochemical methods, the relationship between BC and fire data in the catchment from 1988 to 2018 was analysed. An archive of burnt areas, mapped using NOAA imagery[26], was used to examine fire extent between 1988 and 2018 in the catchments (Fig. 3a–c), and these can be usefully divided into the less intense early dry season (EDS: January–June) fires and the more intense and potentially more destructive late dry season (LDS: July–December) fires[27]. EDS fires include managed burn regimes by government and local indigenous authorities to reduce ground fuel loads intended to prevent more severe fires later in the year. EDS-managed burns were implemented in 2011 and are attributed with reducing the LDS wildfires by about half although the total area burnt on average remained the same[28]. This can be reflected by the relative changes of EDS and LDS areas before and after 2011 (Fig. 3a–c): increased EDS fires and decreased LDS fires.

EDS lower temperature managed burning results in less complete combustion and thus produces more BC. Burnt areas during the EDS showed an obvious increase during 2011–2017, corresponding to increased BC in the three cores after 2010 (Fig. 2c, d, g). Moreover, during the EDS, high rainfall during January to March and southward, offshore winds from March to June would readily transport the emission and ash of wildfires from the land to the sea (Fig. 3d). In contrast, northerly, onshore winds and dry weather dominate during the LDS, which can facilitate fire products to remain inland and reduce their contribution to the sea. This helps to explain the low BC contents in the sediments at the peak of the LDS fire around 2005 (Figs. 2 and 3). These findings demonstrate the effectiveness of BC as a proxy of wildfire frequency and/or intensity over this period and suggest that the impact of EDS fires on the ocean may be more important than that of LDS fires.

In order to assess the broader spatial effect of this phenomenon, we further analysed the relationships between satellite Chl-a concentrations from 2003 to 2017 in the Kimberley coast (total sites: 94,190) and the contents of BC and BSi in Core 200 (2003–2017) (Fig. 4). Chl-a was positively correlated with BC at 34.4% of the sites (Fig. 4a) and with BSi at 54.0% of the sites (Fig. 4b). These results suggest that wildfires may have an extensive spatial effect on MPP in the Kimberley coast.

## Discussion

The effects of increasing wildfires on biogeochemical cycling and primary production in terrestrial ecosystems have attracted great attention[29], but are barely taken into account in marine ecosystems. The results from geochemical reconstruction and observational data in the Kimberley coast indicated that wildfire plays a role in MPP enhancement during the pIOD phase. North-western Australia is one of the world's tropical cyclone hot spots[30]. Research investigating MPP increases in northern Australia has mainly focussed on the contribution of intensified tropical cyclones and rainfall forced by rising temperatures during ENSO years[22,31,32]. Modelling results have suggested that tropical cyclone-induced phytoplankton blooms in north-western Australia could contribute to 20% of annual primary production[11].

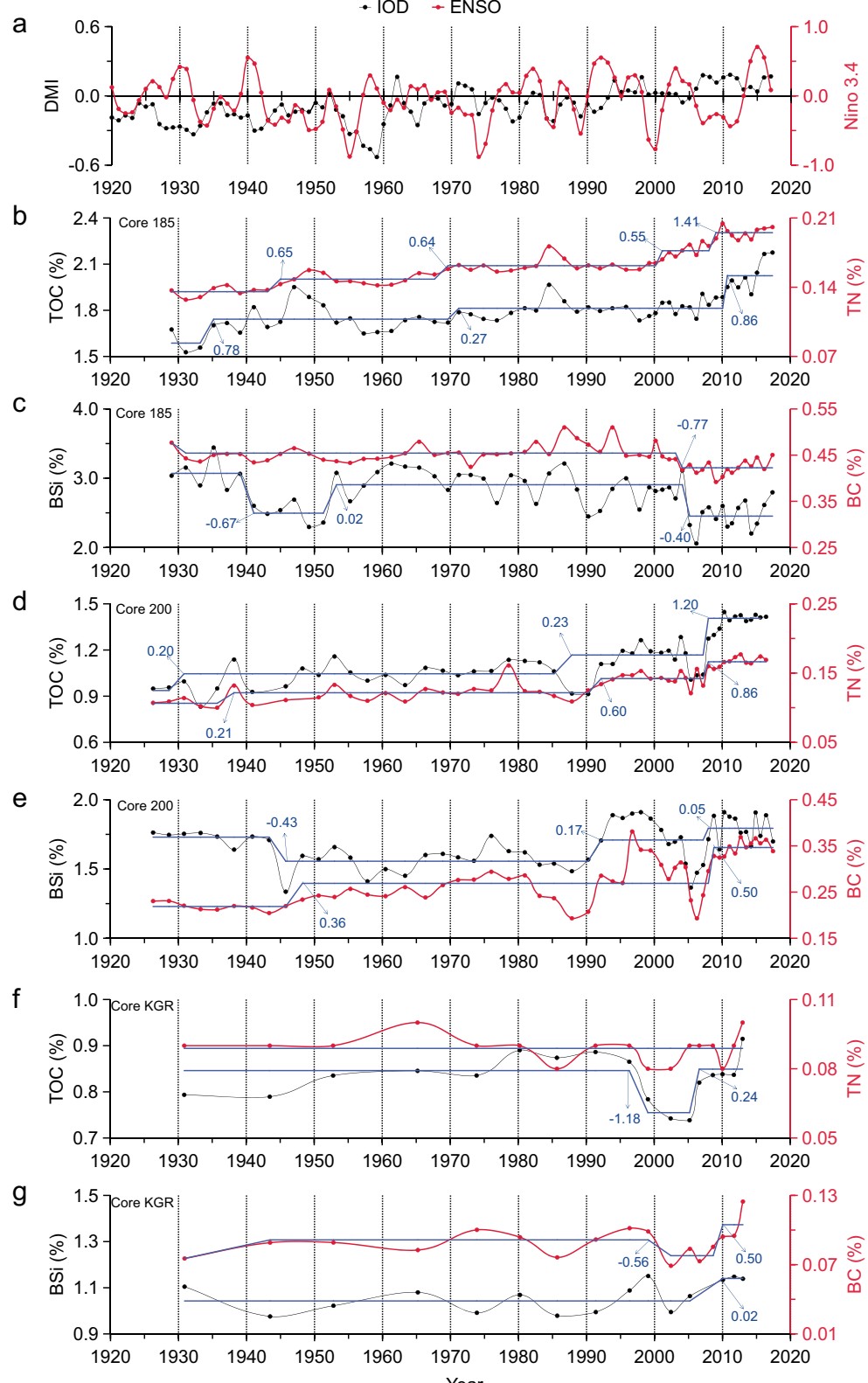

**Fig. 2 The profiles to illustrate the variations of El Niño Southern Oscillation (ENSO), the Indian Ocean Dipole (IOD), and multiple geochemical proxies in the three sediment cores from the 1920s to 2010s. a** Low-frequency signals of dipole mode index (DMI) and Niño 3.4 index showing the variations of IOD and ENSO, respectively. **b** Core 185: total organic carbon (TOC) and total nitrogen (TN). **c** Core 185: biogenic silicate (BSi) and black carbon (BC). **d** Core 200: TOC and TN. **e** Core 200: BSi and BC. **f** Core KGR: TOC and TN. **g** Core KGR: BSi and BC. Blue lines represent the shift changes assessed by sequential *t* test analysis of regime shift and numbers on lines were regime shift index (RSI) to show shifting magnitude.

**Table 1 Pearson correlations (two sided) between each pair of variables in the three cores during the entire time series (ETS: 1926–2017) and the phases of positive Indian Ocean Dipole (pIOD) dominance (1991–2017) and negative Indian Ocean Dipole (nIOD) dominance (1926–1990), respectively (n: number for statistical analysis; r: correlation coefficient; p: the level of significance; significant: p < 0.05; highly significant: p < 0.01; no significant: p > 0.05; bold numbers indicate a significant correlation between two variables).**

| Variables | | Core 185: BSi/BC | | | Core 200: BSi/BC | | | Core KGR: BSi/BC | | |
|---|---|---|---|---|---|---|---|---|---|---|
| | | ETS (n = 56) | pIOD (n = 25) | nIOD (n = 31) | ETS (n = 54) | pIOD (n = 28) | nIOD (n = 26) | ETS (n = 17) | pIOD (n = 10) | nIOD (n = 7) |
| TOC | r | **−0.42**/−0.20 | −0.08/−0.24 | −0.30/0.28 | **0.56**/**0.88** | **0.57**/**0.84** | −0.13/**0.56** | 0.10/**0.55** | 0.24/**0.67** | 0.00/0.04 |
| | p | **<0.01**/>0.05 | >0.05/>0.05 | >0.05/>0.05 | **<0.01**/**<0.01** | **<0.01**/**<0.01** | >0.05/**<0.01** | >0.05/**<0.05** | >0.05/**<0.05** | >0.05/>0.05 |
| TN | r | **−0.48**/**−0.39** | −0.32/**−0.63** | −0.04/0.35 | **0.50**/**0.80** | **0.48**/**0.67** | −0.05/**0.56** | 0.16/0.38 | 0.13/0.48 | 0.56/0.00 |
| | p | **<0.01**/**<0.01** | >0.05/**<0.01** | >0.05/>0.05 | **<0.01**/**<0.01** | **<0.05**/**<0.01** | >0.05/**<0.05** | >0.05/>0.05 | >0.05/>0.05 | >0.05/>0.05 |
| BSi | r | −/**0.41** | −/**0.41** | −/0.19 | −/**0.61** | −/**0.68** | −/0.10 | −/0.47 | −/**0.71** | −/−0.46 |
| | p | −/**<0.01** | −/**<0.05** | −/>0.05 | −/**<0.01** | −/**<0.01** | −/>0.05 | −/>0.05 | −/**<0.05** | −/>0.05 |

The climate data from the present study indicated that an increase in wildfire, sea surface temperature (SST), tropical cyclone frequency, and rainfall are either synchronised or lagged during the pIOD phase (Fig. 2 and Supplementary Fig. 3). Therefore, it is necessary to assess their relative importance in explaining the variance in BSi over time.

Linear Fixed Effect Models and multiple linear regression were applied for the data of BC, SST, TCF, and rainfall in explaining the annual variance of BSi during ETS, pIOD, and nIOD, respectively (Table 2). Model results from cores 185 and 200 showed the positive significance of BC for BSi during ETS; in contrast, the significance of TCF and rainfall for BSi was lower than BC and only appear in Core 200 (Table 2). Multiple linear regression for cores 185 and 200 further showed that BC can explain 36.8 and 8.2% of the variance in annual BSi in the nIOD phase, respectively, but its importance increased to 47.6 and 62% in the pIOD phase, respectively (Table 2). In contrast, SST, tropical cyclones, and rainfall can explain more about the variance in annual BSi in the nIOD phase (Table 2). The result is consistent with the model evaluations in the eastern Indian Ocean between 1997 and 2009, which showed that the atmospheric deposition of nutrients from wildfire accounted for a higher proportion of MPP increase than that of wind-driven nutrient upwelling in pIOD phases[33].

Although the importance of wildfires to MPP in the pIOD phase far exceeds our expectations, the phenomenon is reasonable to some extent. Atmospheric and river transport of macro-nutrients (e.g. nitrogen) and bio-essential metals (e.g. iron) produced by wildfire emission have been considered important contributors to marine primary productivity[16,17]. Albeit the contribution of wildfire elements to the ocean biogeochemical cycle remains unclear, limited observations have indicated their significance. For example, atmospheric nitrogen deposition during the 2006 Indonesian wildfires was three to eight times higher than during non-fire periods and supported the observation of continuously increasing MPP in Sumatra[34]. More recently, anomalously widespread phytoplankton blooms were observed from December 2019 to March 2020 in the Southern Ocean downwind of Australia, attributed to aerosol transport of 2019–2020 Australia wildfires, and high iron contents were observed in the aerosol samples[35]. Iron is an important trace element stimulating phytoplankton growth in oligotrophic oceans[36]. Wildfire is also a major source of soluble iron into the ocean via atmospheric aerosols and river input[37]. In this study, iron and potassium (a typical element in ash) contents in core 200 were measured and they showed distinct increases during pIOD phase (Supplementary Fig. 5). Although the bioavailability of iron and potassium can reduce their significance correlated with BC in the ocean, their synchronous increase in the pIOD phase and positive correlation (Iron: $r = 0.55$, $p < 0.01$; potassium: $r = 0.72$, $p < 0.01$) with BC (Supplementary Fig. 5) indicates a connection with wildfire occurrences.

The results highlight the need, when strong ENSO conditions coincide with pIOD phase, to consider the contribution of wild-fire to the functioning of oligotrophic tropical oceans, not just the role of physical mixing mechanisms (e.g. upwelling, tropical cyclones). It should be noted that the drying effect of the easterly shift in equatorial trade winds induced by pIOD can promote fire conditions during EDS (January–June) and have an important impact on the ocean. The estimation of atmospheric iron deposition shows that the contribution of iron from wildfire is much higher than that of dust iron, particularly in the equatorial Pacific, which has a significant impact on MPP[37]. Thus, it is necessary to strengthen observations of atmospheric deposition and riverine input for estimating the flux of wildfire elements transported to the ocean. Such understanding will provide further

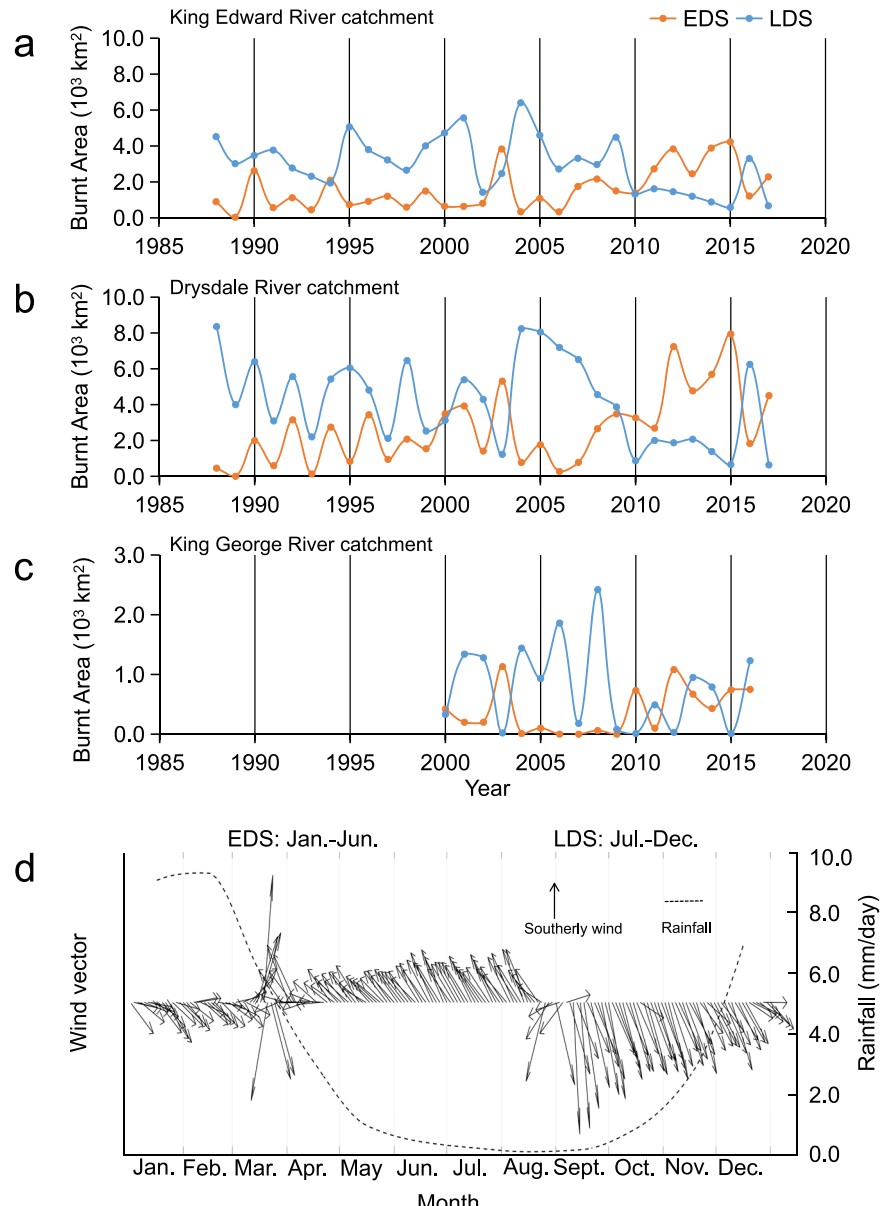

**Fig. 3 Burnt areas of early dry season (EDS: January–June) and late dry season (LDS: July–December) in the catchment of King Edward, Drysdale, and King George Rivers from 1988 to 2017 and seasonal patterns of wind and rainfall in the catchment. a** EDS and LDS fires in the catchment of King Edward River. **b** EDS and LDS fires in the catchment of Drysdale River. **c** EDS and LDS fires in the catchment of King George River. **d** Climatological monthly wind vector and rainfall during EDS and LDS periods.

knowledge on the asymmetry of phytoplankton community responses to climate change in the northern and southern hemispheres.

## Methods

**Core collection and chronological analyses.** The three sediment cores (cores 185, 200, and KGR) used in this study were collected from the northern Kimberley coast (14°02′ S, 126°35′ E; 13°53′ S, 126°45′ E; 13°55′ S, 127°19′ E) (Fig. 1b), using a vibrating head corer (Specialty Devices, Texas, US). Core 185 (=1.31 m long) was extracted near to the King Edward River mouth in 20 m water depth, core 200 (=1.35 m long) was sampled near the Drysdale River mouth in 14 m water depth, and core KGR was sampled near the King George River mouth in 11.8 m water depth (Fig. 1b). The cores were sliced every 0.5 cm along the upper 10 cm, and the rest of the cores were sectioned at 1 cm-thick intervals. All samples were stored in a freezer at −20 °C before dating and chemical analyses.

**Chronological analysis.** The concentration profiles of $^{210}$Pb in the cores were determined by measuring its granddaughter $^{210}$Po radioactive equilibrium using alpha spectrometry[38] at Edith Cowan University, Australia. The concentration of excess $^{210}$Pb used to obtain the age models was determined as the difference between the total $^{210}$Pb and $^{226}$Ra ($^{210}$Pb$_{supported}$). The concentrations of $^{226}$Ra were determined for selected samples along the cores through gamma spectrometry: calibrated geometries in HPGe detectors (CANBERRA, Mod. SAGe Well) were used to measure the decay product of $^{226}$Ra, $^{214}$Pb, at 295 and 352 keV. These concentrations agreed with the total $^{210}$Pb concentrations at depths below the excess $^{210}$Pb horizons (Supplementary Fig. 4). For Core 185 and 200, the Constant Flux: Constant Sedimentation (CF:CS) model[39] was applied below the mixed layer for the 14 to 28 cm sediment core sections, obtaining an average sedimentation rate of 0.247 ± 0.018 g cm$^{-2}$ yr$^{-1}$ for core 185, and 0.338 ± 0.017 g cm$^{-2}$ yr$^{-1}$ for core 200. For Core KGR, the Constant Rate of Supply (CRS) model[40] was applied, obtaining sedimentation rates varying from 0.03 to 0.20 g cm$^{-2}$ yr$^{-1}$.

**Geochemical analysis.** Grain sizes were measured to understand their impact on organic matter preservation, using a Malvern Mastersizer 2000F Laser Particle Sizer. Samples were pre-treated using 10% $H_2O_2$ and 10% HCl to remove organic matter and carbonate, respectively. Then, samples were dispersed in a 0.05% $(NaPO_3)_6$ solution to separate particles for measurement.

The TOC and TN contents in the sediment core were measured using an elemental analyser (FlashSmart NC Soil, Thermo Scientific). Freeze-dried sediment

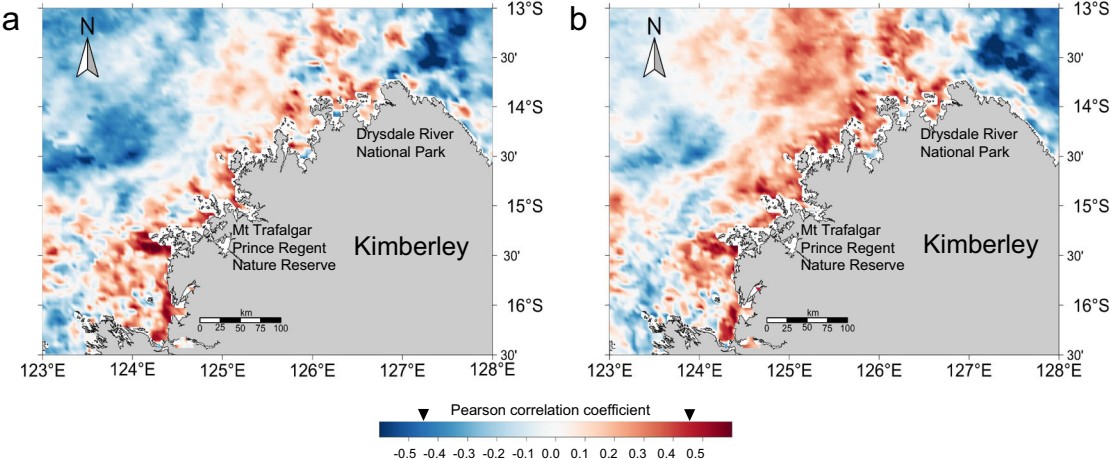

**Fig. 4 Spatial distribution of Pearson correlation coefficients between Chl-a, BC, and BSi during 2003–2017. a** Chl-a and BC. **b** Chl-a and BSi. (two-sided analysis, $n = 17$; Inverted black triangles on the colour bar showing significance ($p < 0.05$) when the correlation coefficients were below $-0.456$ and over $0.456$).

**Table 2 Relative importance in predicting marine phytoplankton production (MPP).**

| | | Estimate Cores 185/200 | Std. error Cores 185/200 | t value Cores 185/200 | p value Cores 185/200 | Relative importance (%) Cores 185/200 |
|---|---|---|---|---|---|---|
| ETS (1926–2017) | Intercept | 3.790/0.908 | 4.333/1.313 | 0.875/0.691 | 0.386/0.493 | — |
| Core 185: $r^2 = 0.217$; $p = 0.012$; | BC | 0.495/0.164 | 0.169/0.033 | 2.924/4.994 | **0.005**/**<0.001** | **90.03**/**87.72** |
| $F_{4,52} = 3.594$ | SST | −0.105/0.007 | 0.144/0.047 | −0.730/0.140 | 0.468/0.889 | 4.76/2.61 |
| Delta_AICc = 14.24 | TCF | 0.026/−0.024 | 0.026/0.011 | 0.990/−2.298 | 0.327/**0.026** | 4.45/6.07 |
| AICcWt = 0.00 | Rainfall | −0.0002/0.0001 | 0.0001/0.00004 | −1.640/2.688 | 0.107/**0.010** | 0.76/3.60 |
| Core 200: $r^2 = 0.489$; $p < 0.001$; | | | | | | |
| $F_{4,49} = 11.71$ | | | | | | |
| Delta_AICc = 0.00 | | | | | | |
| AICcWt = 1.00 | | | | | | |
| pIOD (1991–2017) | Intercept | 5.264/3.308 | 6.648/2.734 | 0.792/1.210 | 0.438/0.239 | — |
| Core 185: $r^2 = 0.289$; $p = 0.129$; | BC | 0.280/0.201 | 0.207/0.048 | 1.351/4.155 | 0.192/**<0.001** | **47.62**/**61.98** |
| $F_{4,20} = 2.031$ | SST | −0.137/−0.082 | 0.220/0.097 | −0.624/−0.841 | 0.539/0.409 | 4.90/1.94 |
| Delta_AICc = 0.00 | TCF | 0.066/0.005 | 0.038/0.288 | 1.769/0.288 | 0.092/0.776 | **45.48**/1.07 |
| AICcWt = 0.98 | Rainfall | $-8.30 \times 10^{-5}$/$8.06 \times 10^{-5}$ | $1.22 \times 10^{-4}$/$5.52 \times 10^{-5}$ | −0.682/1.461 | 0.503/0.158 | 1.99/**35.01** |
| Core 200: $r^2 = 0.543$; $p < 0.001$; | | | | | | |
| $F_{4,23} = 6.821$ | | | | | | |
| Delta_AICc = 41.83 | | | | | | |
| AICcWt = 0.00 | | | | | | |
| nIOD (1926–1990) | Intercept | −2.142/3.527 | 5.855/2.061 | −0.366/1.711 | 0.717/0.101 | — |
| Core 185: $r^2 = 0.038$; $p = 0.890$; | BC | 0.159/−0.016 | 0.298/0.108 | 0.532/−0.152 | 0.599/0.881 | **36.78**/8.24 |
| $F_{4,28} = 0.277$ | SST | 0.153/−0.072 | 0.205/0.069 | 0.749/−1.040 | 0.460/0.310 | **46.10**/7.54 |
| Delta_AICc = 11.02 | TCF | −0.002/−0.027 | 0.035/0.017 | −0.065/−1.576 | 0.949/0.129 | 10.62/**61.69** |
| AICcWt = 0.00 | Rainfall | $-3.26 \times 10^{-5}$/$1.51 \times 10^{-4}$ | $1.64 \times 10^{-4}$/$7.65 \times 10^{-5}$ | −0.199/1.974 | 0.844/0.061 | 6.49/**22.54** |
| Core 200: $r^2 = 0.298$; $p = 0.087$ | | | | | | |
| $F_{4,22} = 2.338$ | | | | | | |
| Delta_AICc = 40.67 | | | | | | |
| AICcWt = 0.00 | | | | | | |

Linear Fixed Effect Models and multiple linear regression, with black carbon (BC), sea surface temperature (SST), tropical cyclones frequency (TCF), and rainfall in explaining the annual variance of biosilicate (BSi) during the entire time series (ETS: 1926–2017), the phases of positive Indian Ocean Dipole (pIOD) dominance (1991–2017) and negative Indian Ocean Dipole (nIOD) dominance (1926–1990), respectively, for Core 185 and Core 200. AIC (Akaike information criterion) model outputs Delta-AICc (difference in AIC score between the best model and the model being compared) and AICcWt (proportion of the total amount of predictive power provided) were used to identify the best model for each core and period (bold numbers indicate the variables contribute greatly to BSi or are significant in Linear Fixed Effect Models).

samples were homogenised by grinding and then acidified with 2 M HCl to remove inorganic carbonate. The acidified samples were then dried in an oven at 60 °C for 1 day, and then washed using Milli-Q water before measurement. Aliquots of approximately 50 mg of the pre-treated samples were used for analysis, and the absolute error for the measurements was <0.3%.

BSi measurements were performed using an INESA-L8 ultraviolet-visible spectrophotometer, according to the Silicon-Molybdenum Blue method[41,42]. Solutions of 10% $H_2O_2$ and 10% HCl were added to freeze-dried and well-milled sediment samples (0.1 g) to remove any organic matter and carbonate. The washed and dried samples were digested with 40 ml 2 M $Na_2CO_3$ at 85 °C for 5 h. At each hour, 0.1 ml of the solution was extracted for absorbance measurements. The volume of each sample was determined to be 10 ml with Milli-Q water, and then add 0.2 ml of HCl, 0.4 ml of ammonium molybdate solution (10.0 g/100 ml), 0.4 ml of ethanedioic acid solution (7.5 g/100 ml), and 0.4 ml of L-Ascorbic acid solution (5.0 g/100 ml) to make the reaction of silicon-molybdenum blue. The absorbance of

molybdenum blue was measured at 660 nm with pure water as a reference, and linear regression was performed for each sample through five absorbance values to estimate the absorbance at $t = 0$ h. The standard curve was drawn according to the known silicon concentration in the solution and the corresponding absorbance value[42]. The concentration of BSi in the sediment was calculated from the concentration of silicon in the solution and the pre-weighted mass of the sediment sample.

Sedimentary BC was quantified using the wet-chemical pre-treatment integrated with the thermal optical reflectance (TOR) method[43], and the TOR method has been proven to effectively discriminate between char and soot[44,45], the two subtypes of BC with different formation mechanisms and physicochemical properties. Prior to the wet-chemical pre-treatment, the sediment samples were firstly thawed, freeze-dried, and homogenised. Approximately 0.20 g of each sample was then digested with HCl/HF to eliminate the inorganic carbonates, metals and metal oxides, and silicates. The remaining residue was filtered through

pre-fired (at 450 °C for 4 h) 47 mm-diameter quartz fibre filters (Whatman) and dried in a constant temperature and humidity chamber following standard methods[45]. BC was separated and quantified on a thermal optical carbon analyser manufactured by Desert Research Institute, Chinese Academy of Sciences. During the analysis, a 0.5 cm$^2$ circular filter punch was placed in an oven. First, the oven was heated in 100% He environment for pyrolysis of organic carbon (defined as $OC_{Pyro}$) and it can be monitored by the 635 nm diode laser. Then the analytical atmosphere was shifted to a mixture of 2% $O_2$ and 98% He, three BC sub-fractions (defined as BC1, BC2, and BC3) will be generated respectively in three temperature stages (580, 740, and 840 °C). All released carbon fractions were oxidised to $CO_2$ with $MnO_2$ as the catalyst and estimated using a non-dispersive infrared detector (NDIR). BC was calculated as the sum of three BC sub-fractions minus the $OC_{Pyro}$ (i.e. BC = BC1 + BC2 + BC3 − $OC_{Pyro}$). For quality assurance and control (QA/QC), a random selection and analysis of 10% of the total filters showed that the relative standard deviation (RSD, %) of the measured BC concentration from different positions within a similar filter was less than 10%, demonstrating the even distribution of the residues onto the filters. In addition, blanks and replicate samples were analysed simultaneously at a frequency of one per ten samples. The blank samples yielded non-detectable BC, and the RSD of replicate analysis averaged 5%.

The thermograms of a series of char and soot reference materials show that char evolves almost exclusively in the element carbon one (EC1) stage and soot in the EC2 and EC3 stages[46,47]. Char content is thus calculated as EC1 minus $OC_{Pyro}$ and soot content as EC2 plus EC3. The high-temperature fossil fuel combustion, such as motor vehicle emissions and industrial coal combustion, show a char/soot ratio of typically less than 1.0, while the relatively low-temperature biomass burning yielded a char/soot ratio significantly higher than 1.0, ranging from 1.2 to ~68 (depending on the fuel type, combustion temperature, air/fuel ratio, and so forth) (Han et al., 2010). The ratios char/soot in the three cores ranges from 6.9 to 22.3 for core 185, from 9.8 to 26.2 for core 200, and from 12.3 to 31.7 for core KGR, with averages of 13.4 ± 3.3, 15.5 ± 3.5, and 21.0 ± 4.4 respectively (Supplementary Fig. 1). Given the Kimberley characteristics, where there is sparse population, low human activity, and massive distance to major cities, together with the relatively higher char/soot ratios, BC in the three cores is dominantly from biomass burning.

Potassium and iron were measured following EPA method 3052[48]. For each sample, 0.25 g dried and homogenised bulk soil was digested at 240 °C using a mixture of concentrated $HNO_3$, HF, and HCl. After total digestion, the samples were diluted with 1% HCl prior to injection into the ICP-MS. Reagent blanks and standard reference materials NIST 2702 (Inorganics in Marine Sediment) and MESS-3 (National Research Council of Canada) were run in parallel to the samples.

**Climate data and satellite Chl-a**. The dipole mode index is used to indicate the IOD phase, which was calculated by the anomalous SST gradient between the western equatorial Indian Ocean (50°E–70°E and 10°S–10°N) and the south-eastern equatorial Indian Ocean (90°E–110°E and 10°S–0°N), based on the ERSST V5 dataset (https://psl.noaa.gov/data/gridded/data.noaa.ersst.v5.html). The Niño 3.4 SST index indicates ENSO, which was calculated by the average SST over the area 5°S–5°N and 170°E–120°W. SST and tropical cyclones from 1920 to 2017 on the Kimberley coast were obtained from the HadISST1 dataset (https://www.metoffice.gov.uk/hadobs/hadisst/) and Australian Tropical Cyclone Database (http://www.bom.gov.au/), respectively. Rainfall data were obtained from two weather stations in Kalumburu (14.30°S, 126.65°E; http://www.bom.gov.au/). The monthly mean wind vector was obtained from NCEP Reanalysis data (https://psl.noaa.gov/data/gridded/data.ncep.reanalysis.html). Fire records with a resolution of 1.1 km pixels were obtained from NOAA Monthly Fire Burnt Areas 1988–2018. Satellite Chl-a data from 2003 to 2017 on the Kimberley coast were MODIS Aqua 4 km Chlorophyll-a level3 products derived from the Asia-Pacific Data Research Center public dataset (http://apdrc.soest.hawaii.edu/data/data.php).

**Statistical analysis**. The correlations between each pair of variables in Table 1 were detected by Pearson correlation analysis, with $r$ representing the correlation coefficient and $p$ representing the level of significance (significant: $p < 0.05$; highly significant: $p < 0.01$; not significant: $p > 0.05$). Linear Fixed Effect Models, using lme4 script in R[49] and multiple regression analysis, were used to quantify the relative importance of each variable in predicting BSi in Table 2. AIC (Akaike information criterion) model selection was used to distinguish among a set of possible models describing the relationships among the variables studied using AICcmodavg script in R[50]. Normality and homoscedasticity of model residuals were estimated visually.

The sequential $t$ test analysis of regime shifts (STARS)[51] was used to examine the shifting time and magnitude of geochemical parameters in Fig. 2. The STARS algorithm converted to VBA for Excel is available from www.BeringClimate.noaa.gov. Briefly, the method uses a $t$ test to determine if sequential records in a time series depart significantly ($p < 0.05$) from mean values in the preceding period (cut-off length set to 10 years). The regime shift index (RSI) represents a cumulative sum of the normalised anomalies, indicating the magnitude of the shift. The spatial distribution of the Pearson correlation coefficients in Fig. 3 was determined based on the annual Chl-a in each grid

(0.01° × 0.01°) between 2003 and 2017, corresponding to the annual contents of BC and BSi in the sediment core.

**Reporting summary**. Further information on research design is available in the Nature Research Reporting Summary linked to this article.

## Data availability
The authors declare that the data supporting the findings of this study are available within the paper and as a Supplementary Information file and may also be requested from D.L. Source data are provided with this paper.

## Code availability
There are no custom codes or custom mathematical algorithms used in this paper.

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

## Acknowledgements

This work was supported by the National Natural Science Foundation of China (42030402, 41876127) and the Western Australian Marine Sciences Institution. O.S. was supported by I + D + i projects RYC2019-027073-I and PIE HOLOCENO 20213AT014 funded by MCIN/AEI/10.13039/501100011033 and FEDER. Funding was provided to P.M. through an Australian Research Council LIEF Project (LE170100219). This work is contributing to the ICTA "Unit of Excellence" (MinECo, MDM2015-0552). The IAEA is grateful for the support provided to its Environment Laboratories by the Government of the Principality of Monaco. Fieldwork was carried out from the RV Solander (KGR) and the PV Worndoom (Napier Broome Bay) and we thank Daryl Moncrieff, Jennifer Munro, Matt de Candia, Michael Hourn, and Ryan Crossing for facilitating and/or assisting with fieldwork. Georgina Pitt and Yueqi Wang assisted with data analysis. Zineng Yuan assisted with core sectioning; Peter Scott analysed the metals' content in the sediment samples. Peter Thompson provided scientific comments for manuscript. All necessary site access and collecting approvals were obtained from the Western Australian Departments of Biodiversity Conservation and Attractions and Primary Industries and Regional Development and Balanggarra traditional owner representatives William Maraltadj Jnr and Bradley Carlton accompanied and assisted in the fieldwork conducted in Napier Broome Bay.

## Author contributions

D.L. formulated concept and undertook the data analysis and writing; C.Z. measured geochemical parameters and carried out statistical analysis; J.K.K. and D.L. carried out field programme and core collection; Y.F. and Y.C. undertook the black carbon analysis; O.S., A.W. and A.S. undertook statistical analysis and interpretation of potassium and iron data; G.P. and J.K. sourced and analysed the burnt area data; P.M. carried out the chronological analyses and Y.D. provided IOD and ENSO data.

## Competing interests

The authors declare no competing interests.

## Additional information

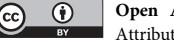

