## [Peer Review File · Nature Communications]

REVIEWER COMMENTS

Reviewer #1 (Remarks to the Author):

This manuscript reported the use of geochemical reconstructions to examine a century-long relationship between wildfire magnitude and marine phytoplankton production (MPP) in a fire-prone region in northern Australia. This research provides meaningful information about the possible impact of wildfire; However, this manuscript needs to be substantially improved before acceptance.

Major issues:

- 1) The manuscript provides an insufficient description of the methods used. For example, how they defined black carbon based on Dr. Han's method, in which both char and soot are calculated. In addition, how exactly marine phytoplankton production was computed?
- 2) The chronological dating was not convincing. What was the depth of the surface mixing layer, and what was the reason the Pb210 had a similar level between 7 to 17 cm? How do you confirm that there was no sediment loss due to resuspension? This is a site not too far away from the river mouth.
- 3) Sedimentary black carbon (BC) level has increased dramatically over the past 150 years due to industrialization (e.g., more fossil fuel usage), which has been found in many sediment cores around the world, even in alpine lakes. Do you have a method to differentiate BC from fossil fuel burning and wildfires?
- 4). Potassium in aerosols can be used for indicating biomass source, but it is not a reliable indicator in waters and sediments because of its high solubility. Other reliable wildfire markers are needed.
- 5). The strong association in one core does not necessarily explain casualty. More sediment cores collected in the upwind and downwind directions of the prevailing wind would be needed to untangle various confounding factors.

Reviewer #2 (Remarks to the Author):

Overall, the manuscript was well written with some minor formatting issues. The premise of the manuscript is also very interesting and novel. My main concern was the statistical analysis (Pearson's correlation). This type of analysis is not suitable to determine which factor is more important and it does not take several factors into account at the same time. Thus, it may lead to the overestimation of importance of factors. My comments are listed below and in the attached manuscript:

1. Line 47: "by rising" should be replaced by "as a result of"
2. Line 186: Formatting issues
3. Line 192: was Pearson's correlation used multiple times? this method may not be the best to determine the relative importance of each variable. A mixed effects model may be more useful in this

instance. Please re-run your analysis with a more robust model.

4. Line 204-205: Again, using Pearson's correlation for individual contents of BC and BSi is not a good method to determine the most important factor. Mixed effects models should be used instead.

5. Line 237: Formatting issues

6. Line 309: This entire section on Statistical analysis should be edited to include other methods (LME or non-parametric analysis).

7. Line 325: "Reference" should be "References"

8. Line 231: Was the data gathered from multiple cores? Or a single core?

Reviewer #3 (Remarks to the Author):

Review of the manuscript "Wildfires enhance phytoplankton production in tropical oceans" by D. Liu et al. submitted for publication in Nature Communications.

This is a sound, well-written and highly relevant study, which deserves publication. The data presented are convincing and I certainly agree with the main conclusions and wider implications presented by the authors. However, a noteworthy caveat needs to be brought to attention. The data obtained from the sediment core are associated with substantial chronological uncertainty, which needs to be discussed when comparing with ENSO and IOD records. The excess ^{210}Pb record, which forms the basis for age determination of the sediment data (Fig. S3) shows uncertainties related to the analytical precision of the ^{226}Ra gamma spectrometry measurements only, whereas the chronological uncertainty of the derived age model is not considered. The latter can be expected to increase with sediment depth to levels that make it impossible to couple details of the sediment records to meteorological data, at least beyond the most recent decade, cf. for example the statement at lines 127-130: "Two shifting points in 1940 and 2010 corresponds to a strong ENSO period (Fig. 2a), while the shifting point in 1995 occurred when the IOD transferred from a negative to a positive phase (Fig. 2b)". This level of precision may be valid for 2010 but certainly not for 1940.

I would like to stress that while this complication severely affects the comparison between sediment records and monitoring data, primarily in the more distant past, it does not compromise the main conclusions of the study as evidenced by the close match between the BC and BSi records during recent decades. If the authors take the chronological uncertainty of the sediment record into account, present an age model with realistic uncertainties, and revise the ms accordingly, the study will warrant publication in Nature Communications.

Response to Reviewers

Reviewer #1 (Remarks to the Author)

This manuscript reported the use of geochemical reconstructions to examine a century-long relationship between wildfire magnitude and marine phytoplankton production (MPP) in a fire-prone region in northern Australia. This research provides meaningful information about the possible impact of wildfire. However, this manuscript needs to be substantially improved before acceptance.

Major issues:

1) The manuscript provides an insufficient description of the methods used. For example, how they defined black carbon based on Dr. Han's method, in which both char and soot are calculated. In addition, how exactly marine phytoplankton production was computed?

Answer: Accepted and implemented. We have defined the black carbon components of soot and char (Lines 95-105) and further clarified the methods description of black carbon (Lines 342-355) and BSi (Lines 305-315) analyses in the revised manuscript.

Further explanation of the methods for black carbon analyses and marine phytoplankton production computations are provided below:

(1). Black carbon: The thermal optical reflectance (TOR) method has been widely used for aerosol black carbon (BC) quantification. To keep the consistency and comparability of BC among different carbon pools, Han et al. (2007a, b) adapted it to quantify BC in sediments and soils after wet chemical pre-treatments. The discrimination between char and soot, which are two subtypes of BC with different

formation mechanism (pyrolysis *versus* gas-to-particle conversion), together with the analyses of physicochemical properties (e.g., particle sizes and therefore transportability), can contribute to understand the nature and geochemistry of BC (Han et al., 2011, 2015; Fang et al., 2018). By carefully examining thermograms of char and soot reference materials, they found that char evolved exclusively in the element carbon 1 (EC1) stage and soot in the EC2 and EC3 stages (Han et al., 2007b). During the analysis, some organic carbon (OC) pyrolyzed (defined as OC_{Pyro}), was further oxidized during a second step involving He/O₂ atmosphere. According to IMPROVE protocol and the aforementioned methods, BC is the sum of char and soot, where char is EC1 minus OC_{Pyro}, and soot is EC2 plus EC3.

References:

- Han, Y. M.; Cao, J. J.; An, Z. S.; Chow, J. C.; Watson, J. G.; Jin, Z. D.; Fung, K.; Liu, S. X. Evaluation of the thermal/optical reflectance method for quantification of elemental carbon in sediments. *Chemosphere* **2007a**, 69 (4), 526-533.
- Han, Y. M.; Cao, J. J.; Chow, J. C.; Watson, J. G.; An, Z. S.; Jin, Z. D.; Fung, K.; Liu, S. X. Evaluation of the thermal/optical reflectance method for discrimination between char- and soot-EC. *Chemosphere* **2007b**, 69 (4), 569-574.
- Han, Y. M.; Cao, J. J.; Yan, B. Z.; Kenna, T. C.; Jin, Z. D.; Cheng, Y.; Chow, J. C.; An, Z. S. Comparison of Elemental Carbon in Lake Sediments Measured by Three Different Methods and 150-Year Pollution History in Eastern China. *Environ. Sci. Technol.* **2011**, 45 (12), 5287-5293.

Han, Y. M.; Bandowe, B. A. M.; Wei, C.; Cao, J. J.; Wilcke, W.; Wang, G. H.; Ni, H. Y.; Jin, Z. D.; An, Z. S.; Yan, B. Z. Stronger association of polycyclic aromatic hydrocarbons with soot than with char in soils and sediments. *Chemosphere* **2015**, *119* (0), 1335-1345.

Fang, Y.; Chen, Y. J.; Lin, T.; Hu, L. M.; Tian, C. G.; Luo, Y. M.; Yang, X.; Li, J.; Zhang, G. Spatiotemporal Trends of Elemental Carbon and Char/Soot Ratios in Five Sediment Cores from Eastern China Marginal Seas: Indicators of Anthropogenic Activities and Transport Patterns. *Environ. Sci. Technol.* **2018**, *52* (17), 9704-9712.

(2). Marine phytoplankton production (MPP): In this study, biosilicate (BSi) and total organic carbon (TOC) were used as MPP proxy. In most oceans, BSi in the sediment mainly originated from diatoms, which are a major component of phytoplankton assemblages and use silicate to synthesize cell walls. Diatom frustules are often used for the reconstruction of marine production in paleoceanography (e.g., Khan et al., 2019; Nelson et al., 1995). The importance of diatom and their contribution for primary production in northern Australia have been emphasised by previous studies (Furnas et al., 2007, 2019; Thompson et al., 2011, 2015). The effectiveness of BSi as an indicator of MPP in northern Australia has been validated using biomarkers (Yuan et al., 2020). Here, we used annual Chl-a data derived from satellite during 2003-2017 in the study area to evaluate its correlation with annual BSi content in the sediment: a positive correlation between Chl-a and BSi suggests that BSi is a reasonable proxy of MPP. In this study, the significantly positive correlations

between TOC and BSi were identified in Core 200 and KGR (Table 1), but not in Core 185. The reason for Core 185 might be related to the changeable grain size and river discharge, because riverine inputs are the most important source of dissolved silicate in the ocean (Tréguer & De La Rocha, 2013).

References:

Khan, M. Z., Feng, Q., Zhang, K. & Guo, W. Biogenic silica and organic carbon fluxes provide evidence of enhanced marine productivity in the Upper Ordovician-Lower Silurian of South China. *Paleogeogra. Paleoclimatol. Paleoecol.* **534**, 109278 (2019).

Nelson, D. M., Treguer, P., Brzezinski, M. A., Leynaert, A. & Queguiner, B. Production and dissolution of biogenic silica in the ocean: revised global estimates, comparison with regional data and relationship to biogenic sedimentation. *Global Biogeochem. Cycles* **9**, 359-372 (1995).

Furnas, M. J. (2007). Intra- seasonal and inter- annual variations in phytoplankton biomass, primary production and bacterial production at north west cape, Western Australia: Links to the 1997- 1998 El Niño event. *Continental Shelf Research*, **27**, 958–980.

Furnas, M. J., & Carpenter, E. J. (2016). Primary production in the tropical continental shelf seas bordering northern Australia. *Continental Shelf Research*, **129**, 33–48.

Thompson, P. A., & Bonham, P. (2011). New insights into the Kimberley phytoplankton and their ecology. *Journal of the Royal Society of Western Australia*, **94**, 161–170.

Thompson, P. A., Bonham, P., Thomson, P., Rochester, W., Doblin, M. A., Waite, A. M., et al. (2015). Climate variability drives plankton community composition changes: The 2010–2011 El Niño to La Niña transition around Australia. *Journal of Plankton Research*, 37(5), 1-19.

Yuan, Z. et al. Phytoplankton responses to climate- induced warming and interdecadal oscillation in North- Western Australia. *Paleoceanogr. Paleoclimatol.* 35, e2019PA003712, (2020).

2) The chronological dating was not convincing. What was the depth of the surface mixing layer, and what was the reason the Pb210 had a similar level between 7 to 17 cm? How do you confirm that there was no sediment loss due to resuspension? This is a site not too far away from the river mouth.

Answer: We agree that the excess ^{210}Pb concentration profile of core 200 presents some deviations from the expected exponential decrease with depth. Using the CF:CS model allows to obtain an average sedimentation rate for the core, where those deviations have a relatively small weight on the model results. In this particular case, they may have been due to local mixing rather than sediment loss due to resuspension or reworking of sediments owing to the presence of excess Pb210 that otherwise would be absent. In order to strengthen the chronological result in this study we have taken the following measures:

We have included a second sediment core ‘**core 185**’ from the bay (see below figure; now included in supplementary as Fig. S3a) for which the age-model is more robust. The sedimentation rate for core 185 was 5.19 ± 0.38 mm/yr, which is similar to

the 4.76 ± 0.25 mm/yr of core 200. The two sediment cores cover the same period of time, and thus we can assess the pIOD and nPOD periods in a reasonably manner.

Artificial radionuclides such as ^{137}Cs can be used in combination with ^{210}Pb to obtain the geochronologies in sediment cores. However, the low inputs of such radionuclides in the study region preclude their use.

Figure S3 The profiles of $^{210}\text{Pb}_{\text{ex}}$ in the two sediment cores (a: core 185; b: core 200).

3) Sedimentary black carbon (BC) level has increased dramatically over the past 150 years due to industrialization (e.g., more fossil fuel usage), which has been found in many sediment cores around the world, even in alpine lakes. Do you have a method to differentiate BC from fossil fuel burning and wildfires?

Answer: We concur, it is an important issue to decipher the provenance of BC

between fossil fuel burning and wildfire sources. The low-temperature pyrolysis

residue of char and high-temperature gas-to-particle condensation of soot can be

distinguished qualitatively by the thermal optical reflectance (TOR) method. The high-temperature fossil fuel combustion, such as motor vehicle emissions and industrial coal combustion, show a char/soot ratio of typically less than 1.0, while the relatively low-temperature biomass burning yielded a char/soot ratio significantly higher than 1.0, ranging from 1.2 to ~68 (depending on the fuel type, combustion temperature, air/fuel ratio, and so forth) (Han et al., 2010). The ratios char/soot in the three cores ranges from 6.9 to 22.3 for core 185, from 9.8 to 26.2 for core 200, and from 12.3 to 31.7 for core KGR, with averages of 13.4 ± 3.3 , 15.5 ± 3.5 , and 21.0 ± 4.4 respectively (Fig. S1). Given the Kimberley characteristics, where there is sparse population, low human activity and massive distance to major cities, together with the relatively higher char/soot ratios, we can confidently assume that BC in the two cores primarily came from biomass burning. The above information can be checked on Line 342-355 and Line 95-105, respectively.

Figure S1 The profiles of Char/Soot ratios in core 185 (a) and core 200 (b).

Reference:

Han, Y. M.; Cao, J. J.; Lee, S. C.; Ho, K. F.; An, Z. S. Different characteristics of char and soot in the atmosphere and their ratio as an indicator for source identification in Xi'an, China. *Atmos. Chem. Phys.* **2010**, *10* (2), 595-607.

4). Potassium in aerosols can be used for indicating biomass source, but it is not a reliable indicator in waters and sediments because of its high solubility. Other reliable wildfire markers are needed.

Answer: In new version, the data of potassium and iron were put into supplementary (See Fig. S5). Potassium and iron are non-conservative elements, they can be used by marine phytoplankton after depositing into the sea, and thus, they are not good enough to be wildfire markers. In contrast to potassium and iron, black carbon (BC) is a conservative element, because they cannot be utilised any phytoplankton or other organisms. Thus, we use BC as wildfire marker in this study. In this study, potassium and iron distinctly increased in the sediment cores after 2010, which is in line with the trend of BC and MPP. Therefore, we use them to indicate the positive effect of ash elements on phytoplankton growth and help to explain why TOC and BSi can increase in the sediment cores. Please see Line 240-246.

5). The strong association in one core does not necessarily explain casualty. More sediment cores collected in the upwind and downwind directions of the prevailing wind would be needed to untangle various confounding factors.

Answer: In revised manuscript, we added the results from another two cores (core 185 and KGR) (Fig. 1b). Black carbon and biosilicates in the three cores (185, 200 and KGR) displayed similar story, BC and BSi displayed and significantly positive correlations during the phase of pIOD dominance (Table 1). The result from Linear Fixed Effect Models and multiple linear regression (Table 2) further confirmed the hypothesis that enhanced wildfire when strong El Niño Southern Oscillation (ENSO) conditions coincided with the positive Indian Ocean Dipole (pIOD) phase enable to promote marine phytoplankton production.

Reviewer #2 (Remarks to the Author)

Overall, the manuscript was well written with some minor formatting issues. The premise of the manuscript is also very interesting and novel. My main concern was the statistical analysis (Pearson's correlation). This type of analysis is not suitable to determine which factor is more important and it does not take several factors into account at the same time. Thus, it may lead to the overestimation of importance of factors. My comments are listed below and in the attached manuscript:

- (1). Line 47: "by rising" should be replaced by "as a result of": **revised**
- (2). Line 186: Formatting issues: **revised**
- (3). Line 192: was Pearson's correlation used multiple times? this method may not be the best to determine the relative importance of each variable. A mixed effects model may be more useful in this instance. Please re-run your analysis with a more robust model.
- (4). Line 204-205: Again, using Pearson's correlation for individual contents of BC and BSi is not a good method to determine the most important factor. Mixed effects models should be used instead.

Answer: We agree with the reviewer and we run a more robust statistical approach, Linear Fixed Effect Models, as suggested, using lme4 script in R (Bates et al., 2015). Mixed effects were not included because only core ID (core 185 vs 200) could be added as a random variable however, we run separate models for each core to strengthen the conclusions derived. We run 8 models in total, 4 models for each core; all models included BSi as the response variable, but different subsets of proxies (predictors) and based on different periods:

Core ID	Period	Model number	Model formula
200	1926 to 2017	1	lm(formula = Bsi ~ BC + SST + TCF + rainfall)
200	1990 to 2017	2	lm(formula = Bsi ~ BC + SST + TCF + rainfall)
200	1926 to 1990	3	lm(formula = Bsi ~ BC + SST + TCF + rainfall)
200	1931 to 2017	4	lm(formula = Bsi ~ K + Fe + C + N + BC + SST + TCF + rainfall)
185	1926 to 2017	5	lm(formula = Bsi ~ BC + SST + TCF + rainfall)
185	1990 to 2017	6	lm(formula = Bsi ~ BC + SST + TCF + rainfall)
185	1927 to 1990	7	lm(formula = Bsi ~ BC + SST + TCF + rainfall)
185	1929 to 2017	8	lm(formula = Bsi ~ K + Fe + C + N + BC + SST + TCF + rainfall)

The results of models 1 to 3 and 5 to 7 fit our previous hypothesis. However, the results of models 4 and 8 do not fit our previous hypothesis. The reason is that K and Fe are non-conservative elements, because they can be consumed by most phytoplankton. Therefore, biological utilization masked the expected correlation of K and Fe with BC and BSi (diatom). In our original Table 1 (Pearson correlations), we showed positive correlations between BSi and BC and K and Fe for some periods, but it was not always significant. However, BC is a conservative element that can't be consumed by any phytoplankton, and showed significance in most of the linear models run. Similarly, C and N are non-conservative elements composed of terrestrial carbon and marine carbon

(e.g., diatom contribution), which could explain the lack of correlation in models 4 and 8.

Based on the rationale above, we kept the original Table 1 showing correlations among variables, and updated Table 2 to show the results of the Linear Fixed Effect Models 1 to 3 and 5 to 7.

References:

Douglas Bates, Martin Maechler, Ben Bolker, Steve Walker (2015). Fitting Linear Mixed-Effects Models Using lme4. *Journal of Statistical Software*, 67(1), 1-48.
doi:10.18637/jss.v067.i01.

5. Line 237: Formatting issues: **revised**

6. Line 309: This entire section on Statistical analysis should be edited to include other methods (LME or non-parametric analysis).

Answer: Yes, we added the method of Linear Fixed Effect Models in revised manuscript, please see Line 385-387.

7. Line 325: “Reference” should be “References”: **revised**

8. Line 231: Was the data gathered from multiple cores? Or a single core?

Answer: The original results were based on one core (ID: 200) collected from the bay. In the revised manuscript, we added the results of another two cores (ID: 185, KGR), one collected from the same bay and the other was from another bay (Fig. 1b). So now

we show the results from three cores (core 185 and 200 and KGR) to strengthen our conclusion.

Reviewer #3 (Remarks to the Author)

Review of the manuscript “Wildfires enhance phytoplankton production in tropical oceans” by D. Liu et al. submitted for publication in Nature Communications. This is a sound, well-written and highly relevant study, which deserves publication. The data presented are convincing and I certainly agree with the main conclusions and wider implications presented by the authors. However, a noteworthy caveat needs to be brought to attention. The data obtained from the sediment core are associated with substantial chronological uncertainty, which needs to be discussed when comparing with ENSO and IOD records. The excess ^{210}Pb record, which forms the basis for age determination of the sediment data (Fig. S3) shows uncertainties related to the analytical precision of the ^{226}Ra gamma spectrometry measurements only, whereas the chronological uncertainty of the derived age model is not considered. The latter can be expected to increase with sediment depth to levels that make it impossible to couple details of the sediment records to meteorological data, at least beyond the most recent decade, cf. for example the statement at lines 127-130: “Two shifting points in 1940 and 2010 corresponds to a strong ENSO period (Fig. 2a), while the shifting point in 1995 occurred when the IOD transferred from a negative to a positive phase (Fig. 2b)”. This level of precision may be valid for 2010 but certainly not for 1940. I would like to stress that while this complication severely affects the comparison between sediment records and monitoring data, primarily in the more distant past, it does not compromise the main conclusions of the study as evidenced by the close match between the BC and BSi records during recent decades. If the authors take the chronological uncertainty of the

sediment record into account, present an age model with realistic uncertainties, and revise the ms accordingly, the study will warrant publication in Nature Communications.

Answer: Thanks for the comments. We totally agree with that the uncertainties associate to the estimated ages need to be taken into account. We made several improvements in the revised manuscript.

(1) The concentrations of excess ^{210}Pb have associated uncertainties obtained from the propagation of errors of the concentrations of total ^{210}Pb and of ^{226}Ra . These uncertainties are then taken into account when obtaining the age model, and thus the estimated average sedimentation rates for the two cores are accompanied by an uncertainty, as well as the calculated ages along the cores. This leads to increasing uncertainty in the age with depth in absolute terms. We have modified the original manuscript focus more on the comparison between sediment records (or compare the correlation of biogenic silica with black carbon and precipitation, tropical cyclones, etc.), taking into account the uncertainties associated to the age estimates. For instance by saying “in the early 1940s” or “before and after 1990s”.

(2) We have included a second sediment core ‘**core 185**’ from the bay (see below figure; now included in supplementary as Fig. S3a) for which the age-model is more robust. The sedimentation rate for core 185 was 5.19 ± 0.38 mm/yr, which is similar to the 4.76 ± 0.25 mm/yr of core 200. The two sediment cores cover the same period of time, and thus we can assess the pIOD and nPOD periods in a reasonably manner.

Artificial radionuclides such as ^{137}Cs can be used in combination with ^{210}Pb to obtain

the geochronologies in sediment cores. However, the low inputs of such radionuclides in the study region preclude their use.

Figure S3 The profiles of $^{210}\text{Pb}_{\text{ex}}$ in the two sediment cores (a: core 185; b: core 200).

(3) To strengthen the observation that the relationship between biogenic silica and black carbon has become more significant during recent decades, we have included the data of a second core (ID: 185), collected from the bay dominated by terrestrial inputs of organic matter (the average carbon/nitrogen ratios of cores 200 and 185 are 8.4 and 11.2, respectively) and a third core (KGR) from the mouth of the King George River. Despite being dominated by organic matter of terrestrial origin, the diatom fraction 'BSi' in the three cores still shows significant correlations with black carbon during *p*IOD phase (Table 1), but not before 1960. It can be visualized in Fig. 2 that the black carbon concentration was continuously decreasing in the late 1980s until a significant downward shift around 2004-2005, a process that was almost simultaneous with the decline in BSi and a downward shift detected at almost the same depth.

REVIEWER COMMENTS

Reviewer #1 (Remarks to the Author):

The authors have addressed all my comments. I recommend the acceptance of this manuscript.

Reviewer #2 (Remarks to the Author):

Overall, the manuscript has been improved according to the comments given in the previous review. My only question is based on the response concerning the number of models used and which model had the lowest AIC (Akaike information criterion) value if AIC criteria was used to determine which model would be used. I understand that several models were used for each core, however, which model fit the data best? The best fitting model can be used to make conclusions for the overall study. As mentioned in the response "However, the results of models 4 and 8 do not fit our previous hypothesis." This may be the case if they are not the best fit for your data. In order for you to make a decision on the best model to use, residual plots and AIC criteria should be used, models with lower AIC values should be prioritized. And AIC values can be inserted into Table 2. Thus, once you include this information in your manuscript, then the best fit model for each core can be found. It was also good to include more than 1 core to support your conclusions as I felt that 1 core was not sufficient.

Reviewer #3 (Remarks to the Author):

In my opinion, the authors have taken my criticism into account to a sufficient degree in their revised version of the manuscript. I would recommend publication.

Response to reviewer

Reviewer #2 (Remarks to the Author):

Overall, the manuscript has been improved according to the comments given in the previous review. My only question is based on the response concerning the number of models used and which model had the lowest AIC (Akaike information criterion) value if AIC criteria was used to determine which model would be used. I understand that several models were used for each core, however, which model fit the data best? The best fitting model can be used to make conclusions for the overall study. As mentioned in the response “However, the results of models 4 and 8 do not fit our previous hypothesis.” This may be the case if they are not the best fit for your data. In order for you to make a decision on the best model to use, residual plots and AIC criteria should be used, models with lower AIC values should be prioritized. And AIC values can be inserted into Table 2. Thus, once you include this information in your manuscript, then the best fit model for each core can be found. It was also good to include more than 1 core to support your conclusions as I felt that 1 core was not sufficient.

Response to reviewer: Thank you for your constructive comment that contributed to improve the manuscript. We run AIC for all models and assessed normality and homoscedasticity of model residuals visually, based the data interpretation on those models with lower Δ_{AICc} and higher $AICcWt$, and added the results in Table 2 as suggested. Dismissing models 4 and 8, as mentioned in our previous response, is

now supported by the AIC outcome: they have high AIC scores and low predictive power.

We also added above information into Table 2 (please see revised Table 2) and Methods section (Line 388-391).

Reference: Mazerolle, M. J. AICcmodavg: Model selection and multimodel inference based on (Q)AIC(c). R package version 2.3-1 (2020).

REVIEWERS' COMMENTS

Reviewer #2 (Remarks to the Author):

The authors have addressed all my comments and concerns. I recommend that this manuscript should be accepted for publication.